# Mealworms as Food Ingredient—Sensory Investigation of a Model System

**DOI:** 10.3390/foods8080319

**Published:** 2019-08-06

**Authors:** Karin Wendin, Viktoria Olsson, Maud Langton

**Affiliations:** 1Department of Food and Meal Science, Kristianstad University, SE-291 88 Kristianstad, Sweden; 2Department of Food Science, University of Copenhagen, DK-1958 Frederiksberg C, Denmark; 3Department of Molecular Sciences, SLU—Swedish University of Agricultural Sciences, SE-750 07 Uppsala, Sweden

**Keywords:** mealworm, *Tenebrio molitor*, insects, sensory, model system

## Abstract

The use of insects as food is a sustainable alternative to meat and as a protein source is fully comparable to meat, fish and soybeans. The next step is to make insects available for use in the more widespread production of food and meals. Sensory attributes are of great importance in being able to increase the understanding of insects as an ingredient in cooking and production. In this pilot study, mealworms were used as the main ingredient in a model system, where the aim was to evaluate the impact on sensory properties of changing particle size, oil/water ratio and salt content of the insects using a factorial design. Twelve different samples were produced according to the factorial design. Further, the effect of adding an antioxidant agent was evaluated. Sensory analysis and instrumental analyses were performed on the samples. Particle size significantly influenced the sensory attributes appearance, odor, taste and texture, but not flavor, whereas salt content affected taste and flavor. The viscosity was affected by the particle size and instrumentally measured color was affected by particle size and oil content. The addition of the antioxidant agent decreased the changes in color, rancidity and separation.

## 1. Introduction

The world’s population is increasing and is expected to reach around 11 billion in 2050 [1], thus increasing pressure on the earth’s resources. FAO (Food and Agriculture Organization of the United Nations) estimates that food production must increase by 70 percent by 2050 to ensure an adequate food supply for the increasing population [1]. However, food production currently has a huge impact on the environment [2] and eating habits need to change, in particular by replacing meat with other protein sources [3,4]. It is necessary to find and use new production sources of protein and other important nutrients. Further, consumers need to be informed about what alternative protein sources that may be available [5]. Common alternatives include the use of plant protein found for example in peas and legumes, another alternative could be to use insect protein. According to the EAT Lancet commission (EAT is the science-based global platform for food system transformation), there is an urgent need for change regarding food and the report concludes that food will be a defining issue of the 21st century [4].

Using insects as food is a sustainable alternative because their production exerts less ecological pressure than that of conventional livestock such as cattle, pigs and poultry as it requires less feed, soil and water [6,7,8]. However, the environmental impact varies greatly between different insect species [9]. Nutritionally, insects are usually fully comparable to other protein sources, such as fish, soybeans or meat e.g., beef, although there is a large variation between species. Insects have been indicated as one of the alternative food sources that could cover the future need for protein and other nutrients [10,11]. Insects contain protein, fat, vitamins and minerals. The nutritional content varies between species, stage of growth and breeding factors [12,13,14]. The protein content is generally high, 20–70% of dry matter, with a high proportion of essential amino acids and good digestibility [15,16]. Insect protein has been shown to be of higher quality than soy protein due to the amino acid composition [17]. The fat content of different insect species varies greatly but most species contain a high percentage of polyunsaturated fatty acids [13,18]. The content of minerals and vitamins varies widely between species, however it should be noted that vitamin B12 can be found in many species, e.g., mealworms [14,19]. 

It is beneficial, both from sustainability and nutritional perspectives, to include insects as a food ingredient [3]. However, a very important criterion for accepting insects as food is that they taste good. The three so-called gateway insects—mealworms, crickets and locusts—are all described as tasty [20]; the flavor of mealworms may be described as nutty with a taste of umami and cereals, cricket resembles popcorn with a hint of chicken and umami, while locust has the flavor of shrimp, nuts and vegetables [21,22]. Insects may be eaten raw, dried or cooked, and also whole, in pieces or milled. They can be cooked in a variety of ways [23], although there is a lack of knowledge regarding how to use insects as an ingredient in everyday cooking. 

Despite all the positive factors, the vast majority of people in Western cultures are reluctant to eat insects and acceptance is generally low. The underlying causes may be attributed to aversion, which can be the result of various sensory signals, such as the ugliness or bad odor of insects, and this is often associated with feelings of anxiety [24]. A negative attitude towards invertebrates, both generally but especially as food, is deeply rooted in Western culture [25]. However, in countries where eating insects is the norm they are seen as a delicacy and a valuable protein source. Knowledge about which species are edible and how they should be prepared is considered important [10].

Internationally, there are more cultures that include insects in their diet than those who do not. It is common to consume insects as food in approximately 120 countries around the world. Further, it is estimated that more than 2000 insect species are edible [26]. Several studies indicate that it is easier for people in western cultures to accept insects as food when the insect ingredient is neutral and less visible or minced than in dishes or products including whole and visible insects [27,28].

The Dutch Council of Affairs describes an increased interest in the consumption of insects and proposes large-scale production in Europe [29]. Dobermann et al. (2017) [10] argue that the economic value of insect production may be higher than that of conventional meat production.

Through a growing awareness of the importance of sustainable food production, the interest in insects as food is increasing. Despite the widespread aversion towards insects as food, interest and acceptance is increasing, not the least since more people consider them as nutritious, sustainable and tasty [20]. The time has come to take the next step by making insects available not only as delicious restaurant food, but also in the more widespread production of food and meals based on insects. The sensory attributes are of great importance in being able to increase the understanding of insects as a main ingredient in cooking and food production.

## 2. Aim

Mealworms were used as the main ingredient in a model system where the aim was to evaluate the impact on sensory properties by changing particle size, oil/water ratio and salt content through the use of factorial design. The effect of an antioxidant agent was also analyzed in samples of differing particle size.

## 3. Material and Methods

### 3.1. Material

Water boiled fresh mealworms (*Tenebrio molitor*, small scale rearer in Sweden) cut/ground into different particle sizes (small: Max 1 mm, medium: Max 3 mm, large: Max 5 mm). The mealworms larvae were fed mainly on oat flakes and carrots, which may affect the flavor. The cut mealworms, sunflower oil (Farm, SR&F, Sweden), water and salt (NaCl, Nordfalks, Sweden) were blended according to a factorial design. The samples were prepared in duplicates. The resulting products were evaluated by descriptive sensory analysis in addition to instrumental measurements of viscosity and color. Nutritional contents were calculated in the software Dietist Net (Kost och Näringsdata, Sweden) by adding values from Finke (2002) [30] to the database. The samples and the ingredients are presented in Table 1 and Table 2. The calculated nutritional values for samples 1–12 are found in Table 3.

The room tempered (19–21 °C) ingredients for samples 1–12 were blended directly before analysis and the samples were then assessed at an ambient temperature of approximately 20 °C (19–21 °C) during a period of 10–40 min after blending.

In addition, Rosemary (Duralox Oxidation Management Blend NM-45, HT, NS, product code: 62-103-03) was added in different amounts to samples 3 and 10. The added amounts were 0%, 0.1% or 0.3% (calculated on weight), Table 2. Five mL portions of each sample were packaged in 15 vacuum-sealed plastic bags and stored in a fridge (+5 °C) for up to 15 days. One bag to be sensorially assessed each day.

### 3.2. Methods

Sensory analysis and instrumental analyses were performed on the samples.

#### 3.2.1. Sensory Analysis

The sensory panel was selected and trained according to the ISO (International Standard Organization) standard 8586-2:2008 [31]. The panel consisted of eight assessors who were first trained in how to perform testing and rate intensity on a numerical scale from 0 to 100, where 0 = no intensity and 100 = highest intensity possible. References and a few selected odor test sample extracts were then presented and assessed in order to reach consensus, i.e., consensus of how to assess the samples and where it was on the intensity scale. Each assessor signed up for participation after being informed about the products and the terms of participation: Voluntary participation, freedom to leave the test without giving a reason and the right to decline to answer specific questions.

Using a slightly modified version of the Flavor Profile Method^®^ [32], the samples in Table 1 were analyzed. The sensory panel was instructed to identify and describe the sensory attributes appearance, odor, taste, flavor and texture. Consensus regarding each attribute should be reached and defined, Table 4. Each attribute was then assessed for each sample by using the intensity scale. Each assessor needed to agree upon the placement of each sample along the intensity axis. The sensory panel members took a break between each testing to refresh their senses. They were instructed to use water and neutral wafers to clean the palate and neutralize the senses. The assessors were informed not to swallow the samples but to spit them out after assessment.

To get an indication of the impact of the added antioxidant, the samples with the added antioxidant agent, rosemary (Table 2), were assessed by two food experts for rancidity, color and texture measured on a five-point scale. The assessments were made each day for 15 days.

#### 3.2.2. Instrumental Measurements 

Instrumental measurements were performed on all twelve samples, see Table 1. Color measurements were performed using a spectrophotometer (Konica Minolta CM-700d) by measuring L*-, a*- and b*-values, where L* corresponds to the lightness, a* to the red-green scale (red is positive and green negative) and b* to the yellow-blue scale (yellow is positive and blue is negative) [33]. Rheology is the study of the deformation and flow of materials and can describe the properties of materials ranging from liquids to elastic solids. One common method is to measure viscosity. In this study, the viscosities of the samples were measured using a rotational viscometer (Myr VR 3000) equipped with a S1 spider.

#### 3.2.3. Presentation and Analysis of Data

Data was analyzed by calculating mean values and standard deviations. Results were correlated by Pearson correlation (Excel, Microsoft Office 2016). Data were also analyzed using the factorial design parameters, linear regression was performed and included all main effects. In order to find trends for interaction effects univariate of analysis of variance was performed with the factorial design parameters two by two. Sensory attributes and instrumental parameters were dependent factors and the factorial design variables were independent (IBM SPSS, version 23). Principal component analysis (PCA; Panel Check V 1.4.2, Nofima, Norway) was performed to give an overview of results.

## 4. Results

Results from the sensory analysis are shown in Figure 1, where the results are divided into the three groups of mealworm particle size: Small, medium and large. An increased particle size increased the yellowness and the perceived coarseness as well as the viscosity and crispness. An increased particle size also resulted in a decreased odor. Increased salt content did, as expected, increase saltiness. It also increased the nutty flavor. Different ratios of oil/water did not seem to impact the sensory properties. 

The results from the instrumental analyses are shown in Table 5 where it can be seen that the measurements could be related to the design factors by results varying in accordance with the design factors. Results from the regression analysis showed that the particle size of the mealworms had a significant impact on the sensory attributes of most of the samples. The results showed that salt content had an impact on the salty taste and nutty flavor, and oil content had an impact on color (Table 6). No significant interaction effects were obtained; however, the interaction of the particle size and oil content was close to being significant, *p* = 0.068, which can also be seen as a trend in Table 5, i.e., viscosity in samples 1–6.

The PCA plot in Figure 2 shows the influence of particle size. It also shows that “particle size” is situated far away from origo along PC1, with this first PC accounting for 69% of the total variation in data. It can, for example, be seen that particle size is positively correlated to viscosity and “Acoarse” and negatively correlated to “Ooat”; the results are supported by the correlations shown in Table 7. Salt content had a positively significant impact on perceived saltiness (“Tsalty”) and Fnutty (Table 6 and Table 7). This can be seen in the PCA plot (Figure 2) where all odd sample numbers (with higher salt content) are located in the upper half together with the Tsalt and salt content, and almost all even sample numbers in the lower half. The ratio of oil/water had no significant impact on perceived sensory parameters (Table 6).

The results obtained from the sensory testing and the instrumental analyses were correlated by Pearson calculations of correlation (Table 7) and these confirm the impact of the particle size on viscosity, color, texture and odor parameters. Further, a high salt content was highly correlated with nutty flavor.

Samples with added rosemary were assessed for rancidity, visual color change and separation. The samples changed slightly and it could be seen that the addition of rosemary had a positive impact on shelf life by slower changes. Samples with larger particles changed color, had a higher rancidity and separated earlier than those with small particle size.

## 5. Discussion

This study makes an important contribution to knowledge about how to formulate the use of insects as a food ingredient. Sensorially appealing products are necessary for their acceptance [20]. From cookbooks we learn that insects may be cooked in a variety of small scale, gastronomic ways [23], but knowledge about how to use insects as an ingredient in industrial food and meal product development is lacking. Ways of rational inclusion of insects in industrial scale product development, to our knowledge, have not systematically been studied. This study evaluated the impact on sensory properties of mealworms in a blend of oil/water and salt where, particle size of the mealworms, oil/water ratio and salt were blended according to a factorial design. The effect of an antioxidant agent was also analyzed in samples differing in particle size. All samples can be considered as high in protein and thereby relevant as a sustainable protein source. The mineral content is high and, according to International Plattform of Insects for Food and Feed -IPIFF (2018) [34], some mineral deficiencies, e.g., iron, may be tackled through the use of insects as food. Contrary to many other sustainable protein sources, e.g., soy beans, insects contain vitamin B12, which may be of importance in future products [14,19].

The flavor of mealworm has been described as nutty with a taste of umami and cereals [21,22]. The result from this study revealed that the nutty flavor might increase with increased salt content, probably due to the polarity of sodium chloride. The results also showed that the particle size of the mealworms had a great impact on appearance and texture, such that an increased particle size increased the yellowness and perceived coarseness. Further, both viscosity and crispness increased. Increased particle size also resulted in decreased odor. We assumed that larger surface area would give larger possibilities for flavor compounds to leak out of the mealworm particle. Increased salt content did not only increase the nutty flavor, but as expected, also increased the saltiness. Different ratios of oil/water did not seem to influence the sensory properties to the same extent as particle size and the addition of salt. Interestingly, the perceived yellowness was not positively correlated to the instrumentally measured b* value. This indicates that humans seem to perceive something different regarding the term “yellow” than that measured by the instrument. All instrumentally measured color values, L*, a* and b* are situated close to origo in Figure 2, indicating little influence on these measures. It can also be observed that higher oil content had a tendency to increase saltiness, as salt is dissolved in water, thus higher oil content results in lower water content e.g., less water for the salt to dissolve in. All samples were prepared using cooked mealworms, thus the protein could already be denatured and less affected by the salt addition, which may not be the case when using unheated mealworms or extracted mealworm protein. 

This model system cannot be regarded as a complete (commercial) food product, since the stability is not high, both in terms of phase separation and in rancidity. With reference to the antioxidative effects of carnosic acid and carnosol [35,36], the addition of rosemary had a significant impact on shelf life in terms of decreased rancidity and color changes. In this study, only two experts evaluated the stability over time and the results are to be regarded as an indication. However, the two assessors might have become too expert in the detection of a rancid flavor to then be able to detect this without bias. After five days there was separation, which has to be considered when converting it to a commercial product. This is a common issue in many food products, however by using hydrocolloids such as starch and solutions this could then be a complete food product [37]. 

In order to make it possible to incorporate insects in food, it is important to better understand what might evoke disgust in relation to insect consumption [38]. Information and education are keys to give objective insight into the connection between food behavior, sustainability and nutrition. Increased knowledge through framed messages might appeal to moral issues such as a person’s value orientation, moral obligation and environmental concerns regarding food choices [38] and overrule factors such as neophobia and disgust that may vary in individuals over the course of a lifetime [39]. Acceptance of edible insects may thus be enhanced by the provision of the right messages. However, sensory and visual aspects of food are important criteria for consumers when deciding on the overall acceptability of a dish [40] and have to be taken into account when developing messages about using insects as food. Our study shows that nutty and oat were the two attributes that described the flavor of the products, which may be regarded as important and appealing attributes in the future development of food products based on mealworms.

## 6. Conclusions

Particle size and salt content significantly influenced the sensory attributes of mealworm samples in a model system. Particle size is therefore an important factor to consider in the development of insects as food products. The addition of an antioxidant agent decreased changes in color, rancidity and separation. This needs to be further considered in future product development.

## Figures and Tables

**Figure 1 foods-08-00319-f001:**
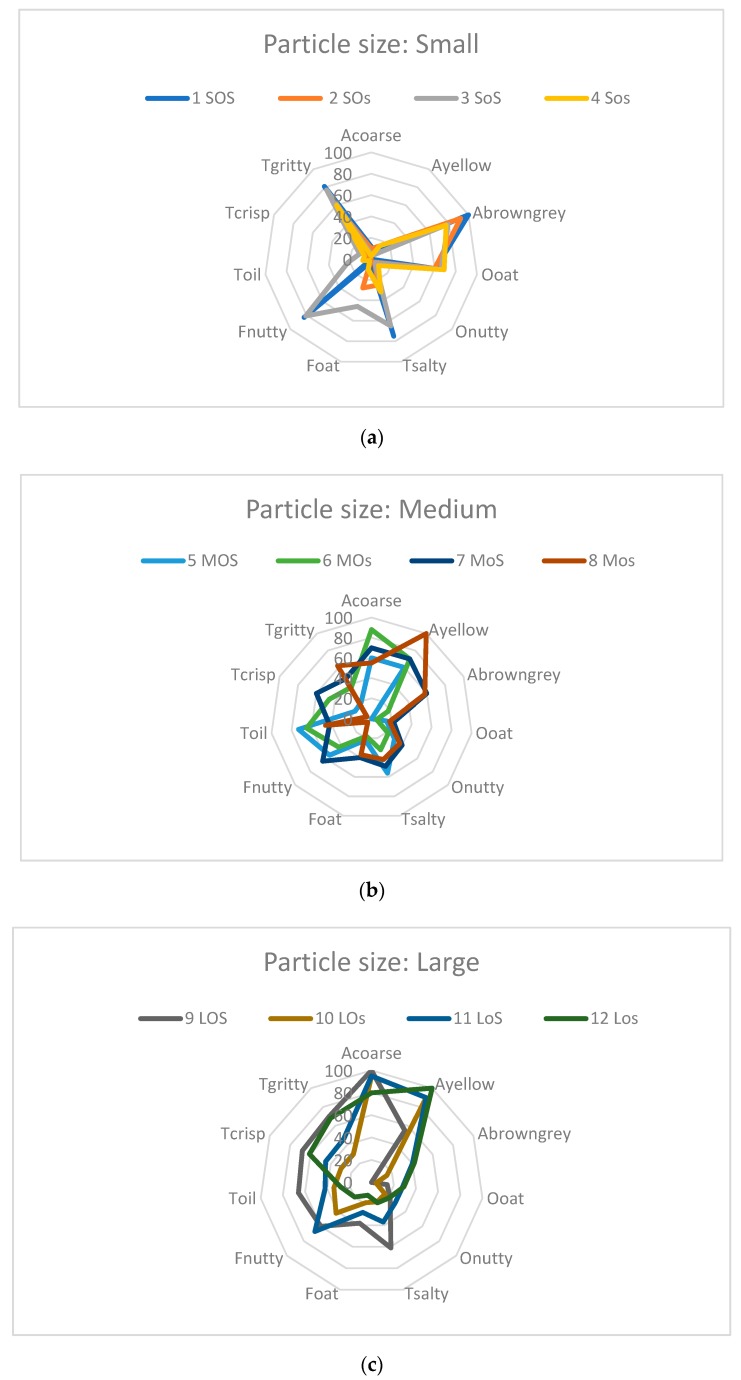
Sensory results displayed in spider plots showing each category of particle size; (**a**) Small particle size, (**b**) Medium particle size, (**c**). Large particle size. Code numbers and identifiers refer to samples in Table 1.

**Figure 2 foods-08-00319-f002:**
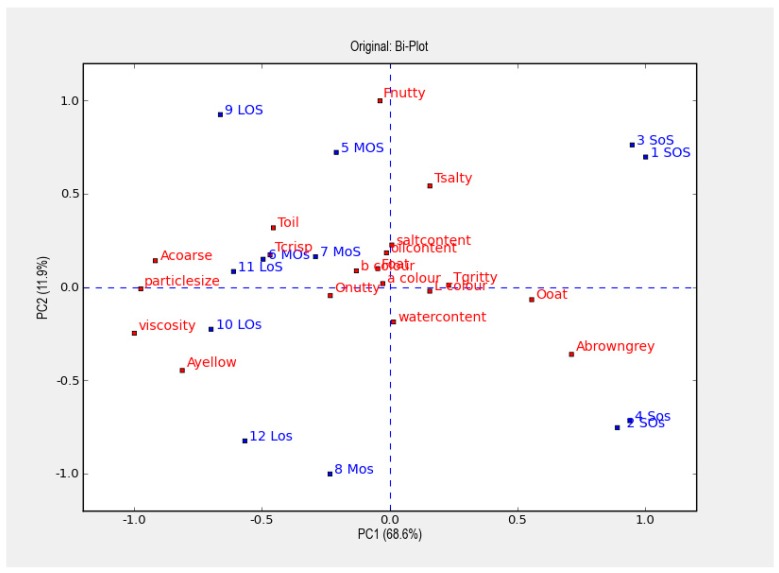
Principal component analysis (PCA) plot giving an overview of all results, visualizing the impact of the design parameters in the factorial design.

**Table 1 foods-08-00319-t001:** Sample preparation with mealworm (50 g) particle size, oil/water ratio of 50 g mixture and NaCl contents.

Sample	Sample Identifier	Particle Size	Oil/Water Ratio (g/g)	NaCl
1	SOS	Small	High/Low	High
2	SOs	Small	High/Low	Low
3	SoS	Small	Low/High	High
4	Sos	Small	Low/High	Low
5	MOS	Medium	High/Low	High
6	MOs	Medium	High/Low	Low
7	MoS	Medium	Low/High	High
8	Mos	Medium	Low/High	Low
9	LOS	Large	High/Low	High
10	LOs	Large	High/Low	Low
11	LoS	Large	Low/High	High
12	Los	Large	Low/High	Low

Samples Identifier: 1st letter refers to particle size (S, M, L), 2nd letter refers to oil/water ratio with o = small amount of oil and O = high amount of oil, 3rd letter refers to NaCl content with S = high amount and s = low amount. Particle size: Small = max 1 mm, medium = max 3 mm, large = max 5 mm. Oil/water ratio: High = 37.5 g, low = 12.5 g. High NaCl: High = 2.0 g, low = 0.5 g.

**Table 2 foods-08-00319-t002:** Samples with added antioxidant agent. Sample with added rosemary where r = 0.1% and R = 0.3% added rosemary.

Sample Identifier	Sample with Added Rosemary	Rosemary (%)
SoS	SoS	0
SoS	SoSr	0.1
SoS	SoSR	0.3
LOs	LOs	0
LOs	LOsr	0.1
LOs	LOsR	0.3

**Table 3 foods-08-00319-t003:** Calculated nutritional content per 100 g sample (calculations on mealworms larvae was based on Finke, 2002 [30] and performed in Software Dietist Net, Kost och Näringsdata, Sweden).

Sample	1	2	3	4	5	6	7	8	9	10	11	12
**Macro Nutrients**												
Protein (g)	9.2	9.3	9.2	9.3	9.2	9.3	9.2	9.3	9.2	9.3	9.2	9.3
Fat (g)	43.4	44.0	18.9	19.1	43.4	44.0	18.9	19.1	43.4	44.0	18.9	19.1
Fiber (g)	3.8	3.8	3.8	3.8	3.8	3.8	3.8	3.8	3.8	3.8	3.8	3.8
Energy (kJ)	1795	1820	888	900	1795	1820	888	900	1795	1820	888	900
Energy (kcal)	429	435	212	215	429	435	212	215	429	435	212	215
**Minerals**												
Calcium (mg)	11.5	9.6	12.4	10.6	11.5	9.6	12.4	10.6	11.5	9.6	12.4	10.6
Iron (mg)	1.0	1.0	1.0	1.0	1.0	1.0	1.0	1.0	1.0	1.0	1.0	1.0
Magnesium (mg)	40.0	40.1	40.2	40.3	40.0	40.1	40.2	40.3	40.0	40.1	40.2	40.3
Sodium (mg)	754	212	724	205	754	212	724	205	754	212	724	205
Zinc (mg)	2.6	2.6	2.6	2.6	2.6	2.6	2.6	2.6	2.6	2.6	2.6	2.6
**Fatty acids (%)**												
Saturated	5.3	5.4	2.8	2.8	5.3	5.4	2.8	2.8	5.3	5.4	2.8	2.8
Monounsaturated	13.5	13.6	6.4	6.5	13.5	13.6	6.4	6.5	13.5	13.6	6.4	6.5
Polyunsaturated	22.5	22.8	8.7	8.8	22.5	22.8	8.7	8.8	22.5	22.8	8.7	8.8
**Vitamins**												
Vitamin A IU	162	164	162	164	162	164	162	164	162	164	162	164
Vitamin D IU	0.3	0.3	0.3	0.3	0.3	0.3	0.3	0.3	0.3	0.3	0.3	0.3
Vitamin E IU	22.4	22.7	7.6	7.7	22.4	22.7	7.6	7.7	22.4	22.7	7.6	7.7
Vitamin C (mg)	2.7	2.7	2.7	2.7	2.7	2.7	2.7	2.7	2.7	2.7	2.7	2.7
Thiamine (mg)	0.1	0.1	0.1	0.1	0.1	0.1	0.1	0.1	0.1	0.1	0.1	0.1
Riboflavin (mg)	0.4	0.4	0.4	0.4	0.4	0.4	0.4	0.4	0.4	0.4	0.4	0.4
Folic acid (µg)	0.1	0.1	0.1	0.1	0.1	0.1	0.1	0.1	0.1	0.1	0.1	0.1
Vitamin B12 (µg)	0.2	0.2	0.2	0.2	0.2	0.2	0.2	0.2	0.2	0.2	0.2	0.2

**Table 4 foods-08-00319-t004:** Sensory attributes and their definitions.

Attribute	Abbreviation	Definition
**Appearance**		
Coarseness	Acoarse	Level of coarseness of mealworm
Yellowness	Ayellow	Intensity of yellow color
Brown-greyish	Abrowngrey	Intensity of brow-grey color
**Odor**		
Oat	Ooat	Odor of oats
Nutty	Onutty	Odor of nuts, no specific type
**Taste/Flavor**		
Salt	Tsalt	Saltiness
Oat	Foat	Flavor of oats
Nuts	Fnutty	Flavor of nuts, no specific type
**Texture (in mouth)**		
Oily	Toil	Fatty, oily sensation
Crispy	Tcrisp	Crisp/crunchy on the first bite
Gritty	Tgritty	Grittiness from peel/shell

**Table 5 foods-08-00319-t005:** Mean values (± SD) of L*, a*, b* color space and viscosity of the sample prepared, *n* = 2.

Sample	L* (m + sd)	a*(m + sd)	b*(m + sd)	Viscosity (mpas; m + sd)
1 SOS	52 + 1.1	4.7 + 0.16	16 + 0.11	1200 + 52
2 SOs	54 + 0.2	4.5 + 0.10	16 + 0.53	1590 + 62
3 SoS	62 + 0.3	3.4 + 0.18	14 + 0.47	400 + 7
4 Sos	60 + 0.2	3.6 + 0.07	14 + 0.14	360 + 14
5 MOS	41 + 1.0	7.1 + 0.28	29 + 1.19	5050 + 1600
6 MOs	37 + 0.2	6.3 + 0.11	24 + 0.04	9700 + 420
7 MoS	45 + 0.2	7.0 + 0.91	25 + 3.80	10,000 + 0
8 Mos	44 + 1.6	5.3 + 0.13	18 + 0.22	10,000 + 0
9 LOS	44 + 0.2	8.7 + 0.95	35 + 4.33	8780 + 630
10 LOs	44 + 0.7	6.0 + 0.09	27 + 0.98	10,000 + 0
11 LoS	46 + 0.2	5.4 + 0.29	22 + 1.63	10,000 + 0
12 Los	46 + 0.1	5.9 + 0.09	25 + 0.15	10,000 + 0

**Table 6 foods-08-00319-t006:** Regression analysis showing the significant (*p* ≤ 0.05) the main effects of the parameters in the factorial design.

	Particle Size	Oil Content	Salt Content
**L***	0.017	ns	ns
**a***	0.008	ns	ns
**b***	0.001	0.037	ns
**viscosity**	0.001	ns	ns
**Acoarse**	0.000	ns	ns
**Ayellow**	0.001	ns	ns
**Abrowngrey**	0.004	ns	ns
**Ooat**	0.004	ns	ns
**Onutty**	ns	ns	ns
**Tsalty**	0.05	ns	0.001
**Foat**	ns	ns	ns
**Fnutty**	ns	ns	0.001
**Toil**	0.05	ns	ns
**Tcrisp**	0.008	ns	ns
**Tgritty**	ns	ns	ns

**Table 7 foods-08-00319-t007:** Pearson correlation coefficients ≥0.7 are given in the table.

	Particle Size	Salt Content	L*	a*	b*	Viscosity	Acoarse	A Yellow	ABrown Grey	Ooat	Onutty	Tsalty
**a***			0.79									
**b***	0.79		0.75	0.96								
**viscosity**	0.87		0.83									
**Acoarse**	0.94		0.84	0.79	0.84	0.93						
**Ayellow**	0.85		0.76			0.94	0.84					
**Abrown Grey**	−0.77		0.76	0.79	0.89							
**Ooat**	−0.79		0.94	0.80	0.81	−0.89	−0.91	−0.83	0.86			
**Onutty**						0.78		0.78		−0.73		
**Tsalty**		0.82										
**Fnutty**		0.86										0.71
**Toil**			0.79	0.79	0.78		0.74		−0.89	−0.84	0.73	
**Tcrisp**	0.77			0.73	0.76	0.73	0.80

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
