# Peer review of "Mealworms as Food Ingredient—Sensory Investigation of a Model System"

_foods, 2019, doi:10.3390/foods8080319_

Round 1
Reviewer 1 Report
This study was well planed and executed. The manuscript is well written, the experimental procedures and statistical analyses are sound, and conclusions are supported by the results.
Only minor corrections are needed. See annotated file for indications on the use of scientific names.

Author Response
This study was well planed and executed. The manuscript is well written, the experimental procedures and statistical analyses are sound, and conclusions are supported by the results.
Only minor corrections are needed. See annotated file for indications on the use of scientific names.
Changement of text style into italic has been done for “Tenebrio Molitor”.
Language has, before first submission, been reviewed by professional and English speaking person

Reviewer 2 Report
Overall evaluation:
This work addresses as one of many already published, the acceptability of insects as food and the authors have chosen food ingredients in the form of a paste, made from mealworms, oil, water, salt and sometimes also a preservative. Good theme, usable in practice. The title states that it is a pilot study - it would be good to describe in detail while the authors see it as the Pilot, if in the "paste", then it should be mentioned in the title of the work. The current title is too general - although it only applies to sensorics and is definitely not a "pilot", mealworm and its sensorics and use in the food industry has been examined many times - see the list of articles for example:
BEDNÁROVÁ, M., BORKOVCOVÁ, M., MLCEK, J., ROP, O. and ZEMAN, L., 2013. Edible insects - Species suitable for entomophagy under condition of Czech Republic. Acta Universitatis Agriculturae et Silviculturae Mendelianae Brunensis, 61(3), pp. 587-593.
BREWER, D., GRAHAM, L., DAVIES, P. and LAJOIE, G., 2002. Identification of novel odorant binding proteins in the beetle tenebrio, molitor, Proceedings 50th ASMS Conference on Mass Spectrometry and Allied Topics 2002, pp. 259-260.
CAPARROS MEGIDO, R., GIERTS, C., BLECKER, C., BROSTAUX, Y., HAUBRUGE, É., ALABI, T. and FRANCIS, F., 2016. Consumer acceptance of insect-based alternative meat products in Western countries. Food Quality and Preference, 52, pp. 237-243.
CHOI, Y.-., KIM, T.-., CHOI, H.-., PARK, J.-., SUNG, J.-., JEON, K.-., PAIK, H.-. and KIM, Y.-., 2017. Optimization of replacing pork meat with yellow worm (Tenebrio molitor L.) for frankfurters. Korean Journal for Food Science of Animal Resources, 37(5), pp. 617-625.
GRAHAM, L.A., BREWER, D., LAJOIE, G. and DAVIES, P.L., 2003. Characterization of a subfamily of beetle odorant-binding proteins found in hemolymph. Molecular & cellular proteomics : MCP, 2(8), pp. 541-549.
KIM, Y., KIM, I. and JEONG, Y., 2019. Quality characteristics of white pan bread added with Tenebrio molitor powder. Journal of the Korean Society of Food Science and Nutrition, 48(2), pp. 253-259.
KRÖNCKE, N., GREBENTEUCH, S., KEIL, C., DEMTRÖDER, S., KROH, L., THÜNEMANN, A.F., BENNING, R. and HAASE, H., 2019. Effect of different drying methods on nutrient quality of the yellow mealworm (Tenebrio molitor L.). Insects, 10(4),.
RONCOLINI, A., MILANOVIĆ, V., CARDINALI, F., OSIMANI, A., GAROFALO, C., SABBATINI, R., CLEMENTI, F., PASQUINI, M., MOZZON, M., FOLIGNI, R., RAFFAELLI, N., ZAMPORLINI, F., MINAZZATO, G., TROMBETTA, M.F., VAN BUITENEN, A., VAN CAMPENHOUT, L. and AQUILANTI, L., 2019. Protein fortification with mealworm (Tenebrio molitor L.) powder: Effect on textural, microbiological, nutritional and sensory features of bread. PLoS ONE, 14(2),.
SOGARI, G., AMATO, M., BIASATO, I., CHIESA, S. and GASCO, L., 2019. The potential role of insects as feed: A multi-perspective review. Animals, 9(4),.
22 Keywords: keywords should be different from the words used in the title, to ensure wider possibilities of searching , „mealworm“ is used in the title, could be replaced by latin name
85 Cooked fresh mealworms (Tenbrio Molitor - should be Tenebrio molitor
94 The ingredients for samples 1-12 were blended directly before analysis and the samples were assessed at an ambient temperature of approximately 20°C. – Value presented this way is non-reproducable. At least minimum and maximum temperature during the processing and storing, before and after blending should be presented. Furthermore, processing interval should be stated, in laboratory samples are often analyzed in minutes or hours after collecting, which is difficult in market practice. There is inevitably a delay and the results may be very different from those gained in laboratory when analysis is done just after grinding.
81 the use of factorial design. The effect of an antioxidant agent was also analysed in samples of differing particle size. - according to Table 2, only samples with small and large particles were analyzed. Why there were no antioxidant analysis for samples with medium sized particles?
121 The samples with added antioxidant agent, rosemary (Table 2), were assessed by 2 food experts – In my humble opinion, 2 people, no matter how experienced they were, were not enough to provide sufficient data for further (statistical) evaluation. Why there were different concentration of the antioxidant agent?
147 Figure 1. Sensory results displayed in spider plots showing each category of particle size. - Figures are very small, it is very difficult to distinguish individual samples. I would suggest to enlarge the figures and use significantly different colors.
180 This study makes an important contribution to knowledge about how to increase the acceptance of insects as a food ingredient. Sensorially appealing products are necessary for their acceptance [20]. From cookbooks we learn that insects may be cooked in a variety of ways [23], but knowledge about how to use insects as an ingredient in commercial and everyday cooking is lacking. – however, this manuscript lacks this information as well, so why is it mentioned?
184 Further, ways of rational inclusion of insects in industrial scale product development, to our knowledge, have not been studied. - There are articles dealing with this topic, as I mentioned above (taken from Scopus).
Author Response
Overall evaluation:
This work addresses as one of many already published, the acceptability of insects as food and the authors have chosen food ingredients in the form of a paste, made from mealworms, oil, water, salt and sometimes also a preservative. Good theme, usable in practice. The title states that it is a pilot study - it would be good to describe in detail while the authors see it as the Pilot, if in the "paste", then it should be mentioned in the title of the work. The current title is too general - although it only applies to sensorics and is definitely not a "pilot", mealworm and its sensorics and use in the food industry has been examined many times - see the list of articles for example:
BEDNÁROVÁ, M., BORKOVCOVÁ, M., MLCEK, J., ROP, O. and ZEMAN, L., 2013. Edible insects - Species suitable for entomophagy under condition of Czech Republic. Acta Universitatis Agriculturae et Silviculturae Mendelianae Brunensis, 61(3), pp. 587-593.
BREWER, D., GRAHAM, L., DAVIES, P. and LAJOIE, G., 2002. Identification of novel odorant binding proteins in the beetle tenebrio, molitor, Proceedings 50th ASMS Conference on Mass Spectrometry and Allied Topics 2002, pp. 259-260.
CAPARROS MEGIDO, R., GIERTS, C., BLECKER, C., BROSTAUX, Y., HAUBRUGE, É., ALABI, T. and FRANCIS, F., 2016. Consumer acceptance of insect-based alternative meat products in Western countries. Food Quality and Preference, 52, pp. 237-243.
CHOI, Y.-., KIM, T.-., CHOI, H.-., PARK, J.-., SUNG, J.-., JEON, K.-., PAIK, H.-. and KIM, Y.-., 2017. Optimization of replacing pork meat with yellow worm (Tenebrio molitor L.) for frankfurters. Korean Journal for Food Science of Animal Resources, 37(5), pp. 617-625.
GRAHAM, L.A., BREWER, D., LAJOIE, G. and DAVIES, P.L., 2003. Characterization of a subfamily of beetle odorant-binding proteins found in hemolymph. Molecular & cellular proteomics : MCP, 2(8), pp. 541-549.
KIM, Y., KIM, I. and JEONG, Y., 2019. Quality characteristics of white pan bread added with Tenebrio molitor powder. Journal of the Korean Society of Food Science and Nutrition, 48(2), pp. 253-259.
KRÖNCKE, N., GREBENTEUCH, S., KEIL, C., DEMTRÖDER, S., KROH, L., THÜNEMANN, A.F., BENNING, R. and HAASE, H., 2019. Effect of different drying methods on nutrient quality of the yellow mealworm (Tenebrio molitor L.). Insects, 10(4),.
RONCOLINI, A., MILANOVIĆ, V., CARDINALI, F., OSIMANI, A., GAROFALO, C., SABBATINI, R., CLEMENTI, F., PASQUINI, M., MOZZON, M., FOLIGNI, R., RAFFAELLI, N., ZAMPORLINI, F., MINAZZATO, G., TROMBETTA, M.F., VAN BUITENEN, A., VAN CAMPENHOUT, L. and AQUILANTI, L., 2019. Protein fortification with mealworm (Tenebrio molitor L.) powder: Effect on textural, microbiological, nutritional and sensory features of bread. PLoS ONE, 14(2),.
SOGARI, G., AMATO, M., BIASATO, I., CHIESA, S. and GASCO, L., 2019. The potential role of insects as feed: A multi-perspective review. Animals, 9(4),.
The title is now changed into “Mealworms as Food Ingredient — Sensory Investigation of a Model System”
22 Keywords: keywords should be different from the words used in the title, to ensure wider possibilities of searching , „mealworm“ is used in the title, could be replaced by latin name
Tenebrio molitor is added to the key words
85 Cooked fresh mealworms (Tenbrio Molitor - should be Tenebrio molitor
Tenebrio Molitor now in Italic
94 The ingredients for samples 1-12 were blended directly before analysis and the samples were assessed at an ambient temperature of approximately 20°C. – Value presented this way is non-reproducable. At least minimum and maximum temperature during the processing and storing, before and after blending should be presented. Furthermore, processing interval should be stated, in laboratory samples are often analyzed in minutes or hours after collecting, which is difficult in market practice. There is inevitably a delay and the results may be very different from those gained in laboratory when analysis is done just after grinding.
Clarifications are made in the text: “The room tempered (19-21°C) ingredients for samples 1-12 were blended directly before analysis and the samples were then assessed at an ambient temperature of approximately 20°C (19-21°C) during a period of 10-40 minutes after blending.”
81 the use of factorial design. The effect of an antioxidant agent was also analysed in samples of differing particle size. - according to Table 2, only samples with small and large particles were analyzed. Why there were no antioxidant analysis for samples with medium sized particles?
The addition of an antoxidative ingredient, added in two different concentrations, was done to give an indication whether it would give an effect or not during a period of 15 days. Thus, only a smaller set of “extreme” samples were used.
121 The samples with added antioxidant agent, rosemary (Table 2), were assessed by 2 food experts – In my humble opinion, 2 people, no matter how experienced they were, were not enough to provide sufficient data for further (statistical) evaluation. Why there were different concentration of the antioxidant agent?
The assessments were done every work day during 15 days by only two assessors to give an indication of the effect of added antioxidative agent. This is an indication and further investigation has to be done. The text is clarified on this point, line 127-129: “To get an indication of the impact of added antixoxidant, the samples with added antioxidant agent, rosemary (Table 2), were assessed by 2 food experts for rancidity, colour and texture measured on a five-point scale. The assessments were made each day for 15 days.” And line 222-223: “In this study, only two experts evaluated the stability over time and the results are to be regarded as an indication.”
147 Figure 1. Sensory results displayed in spider plots showing each category of particle size. - Figures are very small, it is very difficult to distinguish individual samples. I would suggest to enlarge the figures and use significantly different colors.
We agree and send this request to the editors of Foods.
180 This study makes an important contribution to knowledge about how to increase the acceptance of insects as a food ingredient. Sensorially appealing products are necessary for their acceptance [20]. From cookbooks we learn that insects may be cooked in a variety of ways [23], but knowledge about how to use insects as an ingredient in commercial and everyday cooking is lacking. – however, this manuscript lacks this information as well, so why is it mentioned?
Our study is based on a food model system. We learn how the different factors impact the sensory attributes. This is something that could be very useful in cooking. We have modified the text in an attempt to better explain this meaning. Line 189-195: “This study makes an important contribution to knowledge about how to formulate the use of insects as a food ingredient. Sensorially appealing products are necessary for their acceptance [20]. From cookbooks we learn that insects may be cooked in a variety of small scale, gastronomic ways [23], but knowledge about how to use insects as an ingredient in industrial food and meal product development is lacking. Ways of rational inclusion of insects in industrial scale product development, to our knowledge, have not systematically been studied.”
184 Further, ways of rational inclusion of insects in industrial scale product development, to our knowledge, have not been studied. - There are articles dealing with this topic, as I mentioned above (taken from Scopus).
The text is changed to indicate the systematic approach in this study.

Reviewer 3 Report
- The authors used a factorial design by varying the size of mealworms, water, sunflower oil and salt to evaluate their impact on the sensory analysis.
-The interest in using insect as food is increasing every year and insect protein could represent in the future a valuable alternative to the conventional proteins sources. However, the method and analysis are not clear. Please see comments bellow:
Introduction:
-Could you define the abbreviations; FAO, EAT …
-L38: “such as meat” could you precise from which animal?
-L44: “Insect protein has been shown to be of higher quality than soy protein” in which nutrients?
-L 45: “all species contain a high percentage of polyunsaturated fatty acids [13, 18].”not all species contained high percentage of PUFA, black solider fly contain small amounts of PUFA, in addition the content of FAs depends on the media used to grow the larvae.
-L47: “vitamin B12 can be found in many species [14, 19]” an example of the species?
-L50: meal worms and in L85 mealworms.
-L60: “negative attitude towards invertebrates, both generally but especially as food,” How generally?
-L63: “Internationally, there are more people who include insects in their diet than those who do not.” could you explain or elaborate?
Material and methods:
-In this study, you used a factorial design with 12 conditions, which is 24+4. I wondered why you didn’t use a full factorial design with 16 conditions (24) by using the average for the oil, water and salt as you did for the size of mealworms (small, medium and large)?
-In the material and methods part, it is not clear and lot of information are missing:
-L80: oil/water ratio, is not clear in Table 1, could you specify ratio of 3 …
-L 84: many information in this part about the material is missing “cooked fresh mealworms” which method you used for the cooking?
What are the conditions of mealworms rearing, temperature, humidity, the substrate used to grow the larvae?
-L87: Could you specify how the meal was prepared for the sensory testing? The four ingredients: mealworms, oil, water and salt were blended and dried or?
-In Table 1, could you change the title “Samples and sample contents. Particle size: small = max 1 mm, medium = max 3 mm, large = 92 max 5 mm.” explain better that is a factorial design with different factors; precise also that it is sunflower oil and put only NaCl and not salt/NaCl.
L89: could you explain more how you calculated the nutritional composition of the 12 samples, in Table 3?
L95: “before analysis and the samples were assessed” for which analysis? “at an ambient temperature of approximately 20°C” for how long?
L97: why did you choose the sample 3 and 10 to add antioxidant?
L99: “One bag was assessed each day.” for which analysis, could you specify?
L106: Could you write the full name for ISO?
L108-109: “References and a few selected odour test sample extracts were then presented and assessed in order to reach consensus” could explain better this sentence? And also could you explain about the rate of intensity and the numerical scale?
L113: the samples in Table 1 were tested by the eight assessors?
L 121: “were assessed by 2 food experts for rancidity, colour and texture measured on a five-point scale” could you precise more about the food experts, how many years? Also was it enough for the statistical analysis with only two assessors?
L132: what do you mean by S1 spider?
L 133-139: the statistical analysis are not clear; could you specify for each measurement (sensory testing, instrumental measurement and the effect of adding the antioxidant), which software? Data were further analyzed using the factorial design parameters?
In table 6 and figure 2, the analysis are not clear; for the regression analysis you used a response surface model included the main effects and interaction effects. Indeed is not possible to use this model because there are enough degrees of freedom to model the factors (particle size, oil content and salt content), for example how you can model the salt content with only 2 levels, it is impossible to compute such a model? In table 6 there are 3 main effects and 3 interactions = 6 coefficients and 3 levels so it is 3-6 =-3 which is impossible to model.
Results:
The results part are missing some information. In this study you used a factorial design and you described only the results on 1 factor (the size of mealworms). Maybe it could be easy to have a paragraph about each factor but also the interactions?
L 141: could you add comma after “…..Figure 1”
L142: could you add mealworms “three groups of mealworms particle size”
The results in Figure 1 are not clear and very small, it is not possible to see any differences. In order to better understand as well, could you add as supplementary table the sensory scores for all the attributes?
Figure 1: could you precise the figure legend; Spider diagram of the sensory results .. Explaining the code color and the number corresponding to the samples (table 1) and the attributes (Table 4).
L 145: is not possible to see the results in Figure 1 “Different ratios of oil/water did not seem to impact the sensory 145 properties.”
L148-149: “The results from the instrumental analyses are shown in Table 5 where it can be seen that the measurements could be related to the design factors.” Could you explain how the results from the instrumental analyses are related to the design factors?
Table 5: sensory scores from instrumental analyses. Also in the legend could you specify; it is a mean value with standard deviation? n= ? Statistical analysis? Explain again L, a and b.
Table 6: the significant the main effects? How the analysis have been done?
L 170: the results obtained from the sensory testing and the instrumental analyses were correlated ..
L173-176: “The samples changed slightly and it could be seen that the addition of rosemary had a positive impact on shelf life.” are these results presented? and how addition of rosemary had a positive impact on shelf life?
Discussion and conclusions:
L188: “All samples can be considered as nutritionally high in protein and thereby relevant as a more sustainable protein source than meat.” Could you reformulate this sentence?
L 189: “The mineral content is high and, according to IPIFF (2018) [34], some mineral deficiencies may be tackled through the use of insects as food.” Could you specify which minerals and in which insect species?
L191: “Contrary to many other sustainable protein sources” which other protein sources?
L197: “Further, both viscosity and crispness increased” in which condition?
L199: “probably due to decreased exposure of the particle surface” to?
L205: “indicating little influence” on?
L206: “probably due to there being less water available for the salt to dissolve in” could you reformulate?
L226: messages?
In the discussion and conclusion part, there are many information missing, the authors didn’t discuss about the effect of each factor and they just list up a number of observations from the literature and without many speculations. Also what is the main results of this factorial design and how this study could increase the acceptance of eating insect?
-There are many space in the paragraph, example L30, L54…
Author Response
Comments and Suggestions for Authors
- The authors used a factorial design by varying the size of mealworms, water, sunflower oil and salt to evaluate their impact on the sensory analysis.
-The interest in using insect as food is increasing every year and insect protein could represent in the future a valuable alternative to the conventional proteins sources. However, the method and analysis are not clear. Please see comments bellow:
Introduction:
-Could you define the abbreviations; FAO, EAT …
Done: Line 27-28: FAO (Food and Agriculture Organization of the United Nations) and Line 34-35: According to the EAT Lancet commission (EAT is the science-based global platform for food system transformation)
-L38: “such as meat” could you precise from which animal?
Says now line 42: such as meat eg beef,…
-L44: “Insect protein has been shown to be of higher quality than soy protein” in which nutrients?
The text is clarified, line47-48: “Insect protein has been shown to be of higher quality than soy protein due to the amino acid composition”
-L 45: “all species contain a high percentage of polyunsaturated fatty acids [13, 18].”not all species contained high percentage of PUFA, black solider fly contain small amounts of PUFA, in addition the content of FAs depends on the media used to grow the larvae.
Corrected line 48: “The fat content of different insect species varies greatly but most species contain…”
-L47: “vitamin B12 can be found in many species [14, 19]” an example of the species?
Example added, line 51,
-L50: meal worms and in L85 mealworms.
Corrected
-L60: “negative attitude towards invertebrates, both generally but especially as food,” How generally?
This is already explained in the two sentences before
-L63: “Internationally, there are more people who include insects in their diet than those who do not.” could you explain or elaborate?
Changed into line 67: “Internationally, there are more cultures who include…”
Material and methods:
-In this study, you used a factorial design with 12 conditions, which is 24+4. I wondered why you didn’t use a full factorial design with 16 conditions (24) by using the average for the oil, water and salt as you did for the size of mealworms (small, medium and large)?
We choosed this approach in order to be as precise as possible.
-In the material and methods part, it is not clear and lot of information are missing:
-L80: oil/water ratio, is not clear in Table 1, could you specify ratio of 3 …
The table 1 text is now clarified: “Samples and sample contents due to factorial design with the varied factors: particle size, oil/water and salt. Particle size: small = max 1 mm, medium = max 3 mm, large = max 5 mm.”
-L 84: many information in this part about the material is missing “cooked fresh mealworms” which method you used for the cooking?
Clarified into Line 89: “Water boiled fresh mealworms….”
What are the conditions of mealworms rearing, temperature, humidity, the substrate used to grow the larvae?
- Thank you for this comment. We realize that the growing conditions can have an effect on the quality of the mealworms, but this effect was not in focus for the current study. The mealworms were farmed in small scale.
-L87: Could you specify how the meal was prepared for the sensory testing? The four ingredients: mealworms, oil, water and salt were blended and dried or?
Written in the text, Line 89-92: “Water boiled fresh mealworms (Tenbrio Molitor, small scale rearer in Sweden) cut/ground into different particle sizes (small: max 1 mm, medium: max 3 mm, large: max 5 mm), sunflower oil (Farm, SR&F, Sweden), water and salt (NaCl, Nordfalks, Sweden) were blended according to a factorial design.”
-In Table 1, could you change the title “Samples and sample contents. Particle size: small = max 1 mm, medium = max 3 mm, large = 92 max 5 mm.” explain better that is a factorial design with different factors; precise also that it is sunflower oil and put only NaCl and not salt/NaCl.
The Table is corrected
L89: could you explain more how you calculated the nutritional composition of the 12 samples, in Table 3?
This has been explain in Material and methods on line 93-94 as “Nutritional contents were calculated using values from Finke (2002)”.
L95: “before analysis and the samples were assessed” for which analysis? “at an ambient temperature of approximately 20°C” for how long?
Clarified, line 98-100: “The room tempered (19-21°C) ingredients for samples 1-12 were blended directly before analysis and the samples were then assessed at an ambient temperature of approximately 20°C (19-21°C) during a period of 10-40 minutes after blending.”
L97: why did you choose the sample 3 and 10 to add antioxidant?
“Extremes” were choosen to give indication of the affect of added antioxidant.
L99: “One bag was assessed each day.” for which analysis, could you specify?
Now the sentence reads, line104-104: One bag to be sensorially assessed …
More information given in paragraph 3.2.1 Sensory analysis
L106: Could you write the full name for ISO?
Done, line 111-112
L108-109: “References and a few selected odour test sample extracts were then presented and assessed in order to reach consensus” could explain better this sentence? And also could you explain about the rate of intensity and the numerical scale?
The text is modified to better explain the procedure line 113-116:”…. intensity on a numerical scale from 0 to 100, where 0 = no intensity and 100 = highest intensity possible. References and a few selected odour test sample extracts were then presented and assessed in order to reach consensus, ie consensus of how to assess the samples and where it on the intensity scale.”
L113: the samples in Table 1 were tested by the eight assessors?
Yes, written in line 119.
L 121: “were assessed by 2 food experts for rancidity, colour and texture measured on a five-point scale” could you precise more about the food experts, how many years? Also was it enough for the statistical analysis with only two assessors?
The text is changed: line 128-130: “To get an indication of the impact of added antixoxidant, the samples with added antioxidant agent, rosemary (Table 2), were assessed by 2 food experts for rancidity, colour and texture measured on a five-point scale. The assessments were made each day for 15 days.”
L132: what do you mean by S1 spider?
The way the samples were measured by a viscosimeter using this specific rotating inner cylinder
L 133-139: the statistical analysis are not clear; could you specify for each measurement (sensory testing, instrumental measurement and the effect of adding the antioxidant), which software? Data were further analyzed using the factorial design parameters?
Text is changed. Softwares are given in the text.
In table 6 and figure 2, the analysis are not clear; for the regression analysis you used a response surface model included the main effects and interaction effects. Indeed is not possible to use this model because there are enough degrees of freedom to model the factors (particle size, oil content and salt content), for example how you can model the salt content with only 2 levels, it is impossible to compute such a model? In table 6 there are 3 main effects and 3 interactions = 6 coefficients and 3 levels so it is 3-6 =-3 which is impossible to model.
We wanted to find possible trends in the data set, to get indication of interaction effects. Thus we analysed two factors at the time, and one interaction effect was found.
Results:
The results part are missing some information. In this study you used a factorial design and you described only the results on 1 factor (the size of mealworms). Maybe it could be easy to have a paragraph about each factor but also the interactions?
Information of all factors is given
L 141: could you add comma after “…..Figure 1”
done
L142: could you add mealworms “three groups of mealworms particle size”
done
The results in Figure 1 are not clear and very small, it is not possible to see any differences. In order to better understand as well, could you add as supplementary table the sensory scores for all the attributes?
Figure to be larger – we send a request to the editor on this
Figure 1: could you precise the figure legend; Spider diagram of the sensory results .. Explaining the code color and the number corresponding to the samples (table 1) and the attributes (Table 4).
Figure legend is changed: “Sensory results displayed in spider plots showing each category of particle size. Code numbers 1-12 refers to samples in Table 1.”
L 145: is not possible to see the results in Figure 1 “Different ratios of oil/water did not seem to impact the sensory 145 properties.”
Figure to be larger – see answer above.
L148-149: “The results from the instrumental analyses are shown in Table 5 where it can be seen that the measurements could be related to the design factors.” Could you explain how the results from the instrumental analyses are related to the design factors?
Clarified line 156-158: “The results from the instrumental analyses are shown in Table 5 where it can be seen that the measurements could be related to the design factors by results varying in accordance with the design factors.”
Table 5: sensory scores from instrumental analyses. Also in the legend could you specify; it is a mean value with standard deviation? n= ? Statistical analysis? Explain again L, a and b.
Done, the table text now reads: “Table 5. Results, mean value and standard deviation from instrumental analyses: colour (L*- a*- and b*values) and viscosity, n=2”
Table 6: the significant the main effects? How the analysis have been done?
This is explained in the material and method-part
L 170: the results obtained from the sensory testing and the instrumental analyses were correlated ..
Corrected, line 180-181: “The results obtained from the sensory testing and the instrumental analyses were correlated”
L173-176: “The samples changed slightly and it could be seen that the addition of rosemary had a positive impact on shelf life.” are these results presented? and how addition of rosemary had a positive impact on shelf life?
Clarified, line 184-187: “Samples with added rosemary were assessed for rancidity, visual colour change and separation. The samples changed slightly and it could be seen that the addition of rosemary had a positive impact on shelf life by slower changes”
Discussion and conclusions:
L188: “All samples can be considered as nutritionally high in protein and thereby relevant as a more sustainable protein source than meat.” Could you reformulate this sentence?
Changed into, line 200-201: “All samples can be considered as high in protein and thereby relevant as a sustainable protein source”
L 189: “The mineral content is high and, according to IPIFF (2018) [34], some mineral deficiencies may be tackled through the use of insects as food.” Could you specify which minerals and in which insect species?
Specified into, line 201-203: “The mineral content is high and, according to IPIFF (2018) [34], some mineral deficiencies, eg iron, may be tackled through the use of insects as food.”
L191: “Contrary to many other sustainable protein sources” which other protein sources?
Clarified, line 203-204: “Contrary to many other sustainable protein sources, eg soy beans, insects contain vitamin B12 which may be of importance in future products.”
L197: “Further, both viscosity and crispness increased” in which condition?
Explained in the previous sentence
L199: “probably due to decreased exposure of the particle surface” to?
The sentence now reads, line 210-211: “We assume that larger surface area would give larger possibilities for flavour compounds to leak out of the mealworm particle”
L205: “indicating little influence” on?
Now reads, line 218: “….indicating little influence on these measures.”
L206: “probably due to there being less water available for the salt to dissolve in” could you reformulate?
Now, line 219-220:” …as salt is dissolve in water, thus higher oil content results in lower water content e.g. less water for the salt to dissolve in.”
L226: messages?
We have now attempted to make the message more clear, line 234-241:. “In order to make it possible to incorporate insects in food, it is important to better understand what might evoke disgust in relation to insect consumption [38]. Information and education are keys to give objective insight into the connection between food behavior, sustainability and nutrition. Increased knowledge through framed messages might appeal to moral issues such as a person’s value orientation, moral obligation and environmental concerns regarding food choices [38] and overrule factors such as neophobia and disgust that may vary in individuals over the course of a lifetime [39]. Acceptance of edible insects may thus be enhanced by the provision of the right messages.”
In the discussion and conclusion part, there are many information missing, the authors didn’t discuss about the effect of each factor and they just list up a number of observations from the literature and without many speculations. Also what is the main results of this factorial design and how this study could increase the acceptance of eating insect?
See above
-There are many space in the paragraph, example L30, L54…
Check

Round 2
Reviewer 3 Report
Comments and Suggestions for Authors
The authors did not explain the experimental design and the method used for the analysis:
-Table 1: The factorial design is not completely correct, as there are 4 conditions and different levels for each condition (it should be the same level for all the conditions);
Size of mealworm with three level: small, medium and large, sunflower with two level (37.5 and 12.5), water with 2 levels (12.5 and 37.5) and NaCl with 2 levels (0.5 and 1.8 or 1.9).
The average for the oil water and NaCl should be included in the factorial design to have a a full factorial design with 16 conditions.
Also the main problem is the presentation and data analysis:
the Authors did not explain how the data were analyzed, the authors reply that software were added but the problem is not the software but how you use the data to analyse.
The authors reported:
"Data was analysed by calculating mean values and standard deviations. Results were correlated by Pearson correlation (Excel, Microsoft Office 2016). Data were also analysed using the factorial design parameters. A response surface model included all main effects and interaction effects. Sensory attributes and instrumental parameters were dependent factors and the factorial design variables were independent (IBM SPSS, version 23). Principal Component Analysis (PCA) (Panel 159 Check V 1.4.2, Nofima, Norway) was performed to give an overview of results."
It is not possible to use this model because there are not enough degrees of freedom in this model. Also where is the response surface? and the interactions?
In Table 3, I asked the authors for more explanations about the calculation and they did not improve, for example:
In Fink , the nutritional values are given for mealworms but both for larvae and adult so which one they use?, also when I tried to calculate the fat how they did? from which paper they took the values for sunflower, lot of information are missing or just explanations.
Also this paper is about insect and we do not have so much information about the stage, size or just some conditions of insect rearing.
Author Response
The authors did not explain the experimental design and the method used for the analysis:
-Table 1: The factorial design is not completely correct, as there are 4 conditions and different levels for each condition (it should be the same level for all the conditions);
Size of mealworm with three level: small, medium and large, sunflower with two level (37.5 and 12.5), water with 2 levels (12.5 and 37.5) and NaCl with 2 levels (0.5 and 1.8 or 1.9).
The average for the oil water and NaCl should be included in the factorial design to have a a full factorial design with 16 conditions.
Thanks for pointing out this. The design is now clearly defined. Table 1 is revised and contains the factors: Particle size (levels: Small, Medium, Large), Oil/Water Ratio (levels: High/Low, Low/High) and NaCl (Levels: High, Low)
Also the main problem is the presentation and data analysis:
the Authors did not explain how the data were analyzed, the authors reply that software were added but the problem is not the software but how you use the data to analyse.
The authors reported:
"Data was analysed by calculating mean values and standard deviations. Results were correlated by Pearson correlation (Excel, Microsoft Office 2016). Data were also analysed using the factorial design parameters. A response surface model included all main effects and interaction effects. Sensory attributes and instrumental parameters were dependent factors and the factorial design variables were independent (IBM SPSS, version 23). Principal Component Analysis (PCA) (Panel 159 Check V 1.4.2, Nofima, Norway) was performed to give an overview of results."
It is not possible to use this model because there are not enough degrees of freedom in this model. Also where is the response surface? and the interactions?
Thank you for clarification. The text is now improved and reads: “…Data were also analysed using the factorial design parameters, linear regression was performed and included all main effects. In order to find trends for interaction effects univariate of analysis of variance was performed with the factorial design parameters two by two. Sensory attributes and instrumental parameters were dependent factors and the factorial design variables were independent (IBM SPSS, version 23)….”
In Table 3, I asked the authors for more explanations about the calculation and they did not improve, for example:
In Fink , the nutritional values are given for mealworms but both for larvae and adult so which one they use?, also when I tried to calculate the fat how they did? from which paper they took the values for sunflower, lot of information are missing or just explanations.
Table 3 header now reads: “Table 3. Calculated nutritional content per 100g sample (Calculations on mealworms larvae was based on Finke, 2002 [30] and performed in Software Dietist Net, Kost och Näringsdata, Sweden).”
Line 94-95 now reads: “Nutritional contents were calculated in the software Dietist Net (Kost och Näringsdata, Sweden) by adding values from Finke (2002) [30] to the data base.”
Also this paper is about insect and we do not have so much information about the stage, size or just some conditions of insect rearing.
We have added information on the rearing and line 90-91 now reads: “The mealworms larvae were fed mainly on oat flakes and carrots which may affect the flavour.”
For my point of view, this paper does not merit publication as the main problem is the experimental design and the data analysis.
